# Lysosomal Fusion: An Efficient Mechanism Increasing Their Sequestration Capacity for Weak Base Drugs without Apparent Lysosomal Biogenesis

**DOI:** 10.3390/biom10010077

**Published:** 2020-01-03

**Authors:** Nikola Skoupa, Petr Dolezel, Petr Mlejnek

**Affiliations:** Department of Anatomy, Faculty of Medicine and Dentistry, Palacky University Olomouc, Hnevotinska 3, 77515 Olomouc, Czech Republic; NikolaSkoupa@seznam.cz (N.S.); p.dolezel@atlas.cz (P.D.)

**Keywords:** tyrosine kinase inhibitors, lysosomal sequestration capacity, lysosomal fusion, K562 cells, Hl-60 cells

## Abstract

Lysosomal sequestration of anticancer therapeutics lowers their cytotoxic potential, reduces drug availability at target sites, and contributes to cancer resistance. Only recently has it been shown that lysosomal sequestration of weak base drugs induces lysosomal biogenesis mediated by activation of transcription factor EB (TFEB) which, in turn, enhances their accumulation capacity, thereby increasing resistance to these drugs. Here, we addressed the question of whether lysosomal biogenesis is the only mechanism that increases lysosomal sequestration capacity. We found that lysosomal sequestration of some tyrosine kinase inhibitors (TKIs), gefitinib (GF) and imatinib (IM), induced expansion of the lysosomal compartment. However, an expression analysis of lysosomal genes, including lysosome-associated membrane proteins 1, 2 (LAMP1, LAMP2), vacuolar ATPase subunit B2 (ATP6V1B2), acid phosphatase (ACP), and galactosidase beta (GLB) controlled by TFEB, did not reveal increased expression. Instead, we found that both studied TKIs, GF and IM, induced lysosomal fusion which was dependent on nicotinic acid adenine dinucleotide phosphate (NAADP) mediated Ca^2+^signaling. A theoretical analysis revealed that lysosomal fusion is sufficient to explain the enlargement of lysosomal sequestration capacity. In conclusion, we demonstrated that extracellular TKIs, GF and IM, induced NAADP/Ca^2+^ mediated lysosomal fusion, leading to enlargement of the lysosomal compartment with significantly increased sequestration capacity for these drugs without apparent lysosomal biogenesis.

## 1. Introduction

Lysosomes were discovered in the 1950s by Christian de Duve [1,2]. These membrane-bound organelles residing in the perinuclear region are found in all eukaryotic cells. Importantly, in some cell types, lysosomes acquire specific functions. For this reason, they are referred to as lysosome-related organelles [3]. Lysosomes are normally spherical and their size and number is cell-type specific. Importantly, the typical size of lysosomes does not exceed 1 µm in most cells [4].

The structure and function of lysosomes are provided by two types of proteins: lysosomal membrane proteins residing in the lysosomal membrane and soluble acid hydrolases residing in the lysosome lumen. Lysosomal membrane proteins include lysosome-associated membrane proteins (LAMP1, LAMP2), which are heavily glycosylated to protect the lysosomal membrane from acidic hydrolases, lysosomal ion channels (e.g., transient receptor potential mucopolin 1 (TRPML1)), and vacuolar H^+^ ATPase maintaining a low pH. Soluble acid hydrolases include proteases (e.g., cathepsines), nucleases, glycosidases, lipases, phospholipases, phosphorylases, and sulfatases [3].

Initially, lysosomes were considered as the terminal degradative compartment for digestion of both extracellular and intracellular material. However, throughout the years, it has become evident that lysosomes are involved in a plethora of cellular processes and that these organelles are indispensable regulators of cell homeostasis. Currently, lysosomes are considered as multiple signal sensing organelles that regulate cell metabolism, proliferation, gene expression, cell stress, cell death, immune response, membrane repair, tumor invasion, and synaptic plasticity [5,6,7,8]. Mutations of the genes coding for lysosomal proteins that impair lysosomal catabolic functions are responsible for a subset of inherited metabolic disorders known as lysosomal storage diseases [9].

In addition, lysosomes are also recognised as mediators of drug resistance [10,11,12]. Owing to the fact that lysosomes have low internal pH (4.5–6) and the pH in the cytosol is close to neutral (7.2), lysosomes accumulate large amounts of hydrophobic weak base drugs. This phenomenon is also referred to as lysosomal drug sequestration and was theoretically explained many years ago [13,14]. It is assumed that lysosomal sequestration simultaneously reduces the drug availability in the target containing compartment and thus, decreases its therapeutic efficiency [11].

Drug resistance associated with altered intracellular drug distribution caused by extensive lysosomal drug accumulation was first described in anthracyclines [15,16,17,18]. Tyrosine kinase inhibitors (TKIs) represent another important group of weak base anticancer drugs which are reported to have reduced cytotoxicity due to their lysosomal accumulation [19,20,21,22,23].

Recently, a similar idea was extended by novel findings [24]. These authors observed that low non-cytotoxic concentrations of lysosomotropic anticancer drugs induce lysosomal biogenesis mediated by activation of transcription factor EB (TFEB), which in turn further increases the lysosomal sequestration of hydrophobic weak base anticancer drugs and thus, enhances resistance against these drugs [24]. TFEB is a key regulator that triggers transcription of genes encoding proteins of the autophagy-lysosome pathway. Lysosomal biogenesis mediated by TFEB can be induced by different stimuli, including starvation, aberrant lysosomal storage, and pharmacological inhibition of the mTOR (mammalian target of rapamycin) complex 1 [25,26].

In this study, we addressed the question of whether lysosomal biogenesis is the only mechanism that contributes to the enlarged lysosomal capacity of cancer cells in response to treatment with weak base anticancer drugs. We observed that lysosomal fusion induced by TKIs, IM and GF, represent an alternative mechanism to lysosomal biogenesis that substantially increases their sequestration capacity.

## 2. Materials and Methods

### 2.1. Chemicals

Gefitinib (Iressa, ZD1839, GF; purity ≥ 99%) was purchased from Selleckchem (Huston, TX, USA). Imatinib mesylate (STI571, Gleevec; IM, purity ≥ 98%) was kindly provided by Novartis (Basel, Switzerland). Bafilomycin A1 (BafA1), selective inhibitor of the vacuolar H^+^-ATPase, was purchased from MedChemExpres (Monmouth Junction, NJ, USA). BABTA-AM (1,2-bis(2-aminophenoxy)ethane-*N*,*N*,*N*′,*N*′-tetraacetic acid tetrakis (acetoxymethyl ester) the cell permeable, selective Ca^2+^ chelator was purchased from Sigma-Aldrich (Saint Louis, MO, USA). Nicotinic acid adenine dinucleotide phosphate (NAADP) antagonist, *trans*-Ned 19, (1*R*,3*S*)-1-[3-[[4-(2-Fluorophenyl)piperazin-1-yl]methyl]-4-methoxyphenyl]-2,3,4,9-tetrahydro-1*H*-pyrido [3,4-*b*] indole-3-carboxylic acid, was purchased from Tocris Bioscience (Abingdon, UK).

### 2.2. Cell Culture

Human chronic myelogenous leukemia K562 cells and human promyelocytic leukemia HL-60 cells were cultured in the RPMI-1640 medium supplemented with a 10% calf foetal serum and antibiotics in 5% CO_2_ atmosphere at 37 °C. Cells were obtained from the European collection of authenticated cell cultures (ECACC; Salisbury, UK).

### 2.3. Assay for Determination of Intracellular IM Levels

Cells (density of 4 × 10^5^/mL) were incubated in the growth medium with appropriate IM concentration in the absence or presence of BafA1 for 6 h in 5% CO_2_ atmosphere at 37 °C. Cell pellets were extracted using ice cold 4% (*w*/*v*) formic acid (FA) in water on their separation from the growth medium by centrifugation through a layer of silicone oil. Cell extracts were clarified by centrifugation (40,000× *g* × 10 min at 4 °C), diluted with extraction solution and analysed by liquid chromatography coupled with a low-energy collision tandem mass spectrometer (LC/MS/MS). Details are given elsewhere [27,28,29].

### 2.4. Assay for Determination of Intracellular GF Levels

Cells (density of 5 × 10^5^/mL) were incubated in the growth medium with appropriate GF concentration in the absence or presence of BafA1 for 6 h in 5% CO_2_ atmosphere at 37 °C. Cell pellets were extracted using ice cold 5% (*w*/*v*) FA in water upon their separation from the growth medium by centrifugation through a layer of silicone oil [27,28]. Cell extracts were clarified by centrifugation (40,000× *g* × 10 min at 4 °C), diluted with extraction solution and analysed or stored at −80 °C.

Quantitative analysis of GF was done using liquid chromatography coupled with a low-energy collision tandem mass spectrometer (LC/MS/MS) during one run. The HPLC system UltiMate 3000 (Dionex, Germering, Germany), a HyperClone BDS C18, 5 µm, 150 × 2.0 mm HPLC column (Phenomenex, Torrance, CA, USA) and a guard C18, 4.0 × 2.0 mm precolumn (Phenomenex, Torrance, CA, USA) were used. The chromatographic parameters were optimized: the binary gradient of mobile phase A (95% methanol in 0.25% formic acid, *v*/*v*) and B (0.25% formic acid in water, *v*/*v*) from 0–4 min (35%→95% of solvent A), from 4–6 min (95% of solvent A), from 6–7 min (95%→35% of solvent A) and from 7–10 min (35% of solvent A); the flow rate at 0.3 mL/min; the sample injection volume at 5 µL. The API 3200 triple quadrupole mass spectrometer (MDS SCIEX, Markham, ON, Canada) with the TurboIonSpray interface in the positive ion mode was applied for quantification of analytes. Ion spray probe parameters were set for standard: needle voltage 5500 V and temperature 400 °C. The mass spectrometer was operated in the multiple-reaction monitoring (MRM) mode. GF was monitored by MRM transition 447 > 128. The mass-dependent parameters were optimized: the collision energy and the declustering potential for GF standard were 46 V and 35 V. Data were acquired using Analyst^®^ software (ver. 1.5.1, AB SCIEX, Markham, ON, Canada).

### 2.5. Calculation of TKIs in Lysosomes

Absolute accumulation of TKIs in lysosomes was calculated as described previously [30,31]. Briefly, the value of intracellular accumulation of particular TKI in the presence of BafA1 was subtracted from the value of intracellular accumulation of particular TKI in the absence of BafA1. Absolute accumulation of TKIs in lysosomes is expressed as molar amount of particular TKI in lysosomes per 10^6^ cells. Relative accumulation of TKIs was calculated as follows. The absolute value of accumulation of particular TKI in lysosomes is divided by the value of intracellular accumulation of TKI. Relative accumulation of TKIs is expressed in percentage.

### 2.6. Western Blot Analysis

Preparation and processing of protein samples were done as described elsewhere [32]. Briefly, cells were washed in ice cold phosphate buffered saline (PBS) and proteins were extracted using lysis buffer (50 mM Tris/HCl buffer pH 8.1 containing 1% NP-40, 150 mM NaCl, 50 mM NaF, 5 mM EDTA, and 5 mM sodium pyrophosphate, supplemented with protease (Roche, Mannheim, Germany) and phosphatase (Sigma-Aldrich, Saint Louis, MO, USA) inhibitor cocktails. Cell extracts were heat denatured in Laemmli buffer (31.25 mM Tris/HCl, pH = 6.8, 10% glycerol, 2% SDS, 5% 2-mercaptoethanol, 0.005% bromphemol blue). Samples (30 µg protein) were separated by SDS-PAGE on 10% gels and transferred onto nitrocellulose membranes.

Lysosomal proteins were analyzed using rabbit monoclonal anti-LAMP1 (D2D11) XP antibody (1:1000), rabbit monoclonal anti-LAMP2 (D5C2P) antibody (1:1000), and rabbit monoclonal anti-ATP6V1B2 (D2F9R) antibody ((1:1000); Cell Signaling Technology, Denvers, MA, USA).

To confirm equal protein loading, immunodetection was performed with the rabbit polyclonal anti-β-actin antibody ((1:2000); Sigma-Aldrich, St. Louis, MO, USA). The signal was detected using a horseradish peroxidase-conjugated secondary antibody ((1:15,000); Dako, Glostrup, Denmark). Products were visualized using an enhanced chemiluminesence (ECL; Amersham, Little Chalfont, UK).

### 2.7. Activity of Lysosomal Hydrolases

Expression of acidic phosphatase (ACP) and β-d-galactosidase (GLP) was determined by measurement of their enzymatic activity in cell lysates [24,31]. Cells were washed in ice cold PBS and lysed in buffer (25 mM HEPES (pH 7.0), 0.5% CHAPS, 0,5% Triton X-100, 0.5 mM dithiothreitol, 2 mM EGTA, and protease cocktail inhibitors) on ice for 30min. Cell extracts were clarified centrifugation (15,000 rpm/15 min/4 °C). Protein concentration was determined by the Bradford method [33]. The enzymatic reaction was initiated by adding cell extract (equivalent of 100 μg protein) to the cell assay buffer (50 mM NaCl, 50 mM citrate–phosphate buffer, pH 4.5) containing appropriate substrate. ACP activity was measured using 1 mM 4-methylumbelliferyl phosphate as a substrate. GLB activity was measured using 1 mM 4-methylumbelliferyl-β-d-galactopyranoside as a substrate. The reaction mixture was incubated at 37 °C for 30 min and then the enzymatic reaction was stopped by adding 50 µL of 1 M Tris buffer, pH 10.7. The relative fluorescence of released 4-methylumbelliferone was monitored at 365/445 nm.

### 2.8. Immunostaining of LAMP1

We used the method described previously [34]. Briefly, cells were washed in ice cold PBS (pH 7.4) and then fixed using 4% paraformaldehyde in PBS for 15 min at laboratory temperature. Fixed cells were washed in ice-cold PBS (three times) and then permeabilised in PBS containing 0.1% Triton X-100 for 15 min. Nonspecific binding of the antibodies was blocked by transferring cells into PBS containing 1% BSA (bovine serum albumin) and 0.1% Tween 20 for 30 min. Cells were further incubated with rabbit monoclonal anti-LAMP1 (D2D11) XP antibody (1:50; Cell Signaling Technology, Denvers, MA, USA) in PBS containing 1% BSA and 0.1% Tween 20 in a humidified chamber overnight at 4 °C. Unbound primary antibody was washed out with PBS (three times). Cell incubation with the secondary antibody diluted in PBS containing 1% BSA and 0.1% Tween 20 proceeded for 1 h at room temperature in the dark. Unbound secondary antibody was washed out with PBS (three times) in the dark. Nuclei of cell samples were stained by DAPI working solution (part of the kit) for 5 min, washed and coverslipped with mounting medium.

Lysosomes were visualized by confocal laser scanning microscopy (Fluorview 1000 unit attached to IX80 microscope; Olympus Czech Group, Prague, Czech Republic). Due to photobleaching of FITC sequential scanning mode was applied, starting with excitation of FITC by a 488 nm line of argon laser and signal detection by a 505–550 nm emission filter, followed by subsequent DAPI excitation by a diode laser at 405 nm and emission at 430–470 nm. Optical sections through a central plane of each cell were performed with the following set-up: oil immersion objective 100×, zoom 2.8×, C.A. 200, image size 1024 × 1024 px, scan 20 µs/px. At the beginning of experiment, the laser intensity was set based on control samples omitting primary antibody. To increase the visibility of signal in all images, the FITC channel was exported with brightness levels limited from 0–4095 to the range of 40–2140.

The ImageJ software (free software provided by NIH-National Institutes of Health; Wayne Rasband) was used for image analysis. Lysosomal morphology was evaluated in at least 250 cells for each treatment.

### 2.9. Statistical Analysis

Data are reported as the means ± S.D. Statistical analyses were performed using SigmaPlot 11.0 software package (Systat Software Inc., San Jose, CA, USA). Statistical significance was determined by a Student’s t-test (when the means of only two groups was to be compared), and one-way ANOVA (when means of more than two groups were to be compared). We used * or # for the significant result (*p* < 0.05) and ** or ## for the very significant result (*p* < 0.01).

## 3. Results

We studied the effect of TKIs on lysosomal capacity in human leukemia K562 and HL-60 cell lines representing models for chronic myeloid (CML) leukemia and acute myeloid leukemia (AML), respectively. At the present time, patients with CML are successfully treated with TKIs (e.g., IM, nilotinib, and dasatinib) and clinical trials are underway to test TKIs for the treatment of AML [35,36]. Given that the cytotoxic effect of weak base drugs can be compromised by lysosomal sequestration [10,11,12], investigating the effect of TKIs on the sequestration capacity of lysosomal compartment in K562 and HL-60 cells is of great importance.

Not surprisingly, we found that GF and IM significantly accumulated in lysosomes of cancer cells (Figure 1). The absolute lysosomal accumulation of GF and IM increased with increasing extracellular concentration without reaching a plateau (Figure 1a,c). Importantly, the relative accumulation of GF and IM in lysosomes also increased with increasing extracellular concentration: the higher the extracellular GF and IM concentration, the greater was the percentage of drug accumulated in lysosomes (Figure 1b,d). This effect was more pronounced for IM (Figure 1c,d). These results suggest that GF and IM induced an enlargement of the lysosomal compartment.

Therefore, we addressed the question of whether the increased lysosomal accumulation capacity of GF and IM was associated with lysosomal biogenesis since such effect was observed for various lysosomotropic drugs [24,37]. However, Western blot analysis of lysosomal marker genes, including *LAMP1*, *LAMP2*, and *ATPGB2* (vacuolar ATPase subunit B2), indicated no change in their expression at protein level (Figure 2 and Figure 3). Similarly, lysosomal enzymes ACP, and GLB did not exhibit increased activity (Figure 4). In contrast, a decrease in ACP and GLB activity was observed for the highest concentrations of GF and IM in HL-60 and particularly in K562 cells (Figure 4). These results indicate that the increased lysosomal sequestration capacity can hardly be attributed to lysosomal biogenesis.

We further observed that both TKIs induced changes in morphology of lysosomes that was more apparent at higher concentrations of GF and IM (Figure 5 and Figure 6). Detailed quantitative analysis of morphological changes in lysosomes of cells with increased lysosomal sequestration capacity indicated that their size significantly increased and simultaneously the number of lysosomes significantly decreased (Figure 5 and Figure 6). Importantly, *trans*-Ned 19 that blocks NAADP-mediated Ca^2+^ release [38] abrogated observed morphological changes (Figure 7 and Figure 8). In line with these results, we further observed that BAPTA-AM, a cell-permeant Ca^2+^ chelator, also prevented the observed morphological changes (Figure 9 and Figure 10). The simplest explanation for these observations was that GF and IM induced NAADP-mediated Ca^2+^ release triggering lysosomal fusion, which led to the increased lysosomal capacity. The theoretical analysis clearly indicates that lysosomal fusion may be the mechanism that could explain the expansion of the lysosomal compartment without lysosomal biogenesis (Figure 11). For example, while the fusion of two identical lysosomes gives a final volume that is 2.8 times larger, the fusion of four identical lysosomes gives a final volume that is eight times larger (Figure 11b).

## 4. Discussion

Sequestration of weak base anticancer drug in lysosomes that lead to reduced drug availability in the target containing compartment, is considered one of the mechanisms that contribute to MDR [11]. Lysosomal biogenesis in response to nontoxic concentration of hydrophobic weak base chemotherapeutics driven by TFEB was a recently described mechanism which increases lysosomal sequestration of hydrophobic weak base drugs, thereby increasing the MDR phenotype [24]. However, it is probably not a universal mechanism that occurs in all types of cancer cells. In our laboratory, we have recently shown that A549, MCF7 and K562 tumor lines exposed to non-toxic sunitinib concentrations for three days, increased significantly their ability to sequester various TKIs by lysosomes without observable lysosomal biogenesis [31].

In this study, we made further progress and showed that in addition to lysosomal biogenesis, there is another, recently identified mechanism, leading to increase in the sequestration capacity of lysosomes in tumor cells. This is lysosomal fusion induced by the treatment of K562 and HL-60 cells with micromolar concentrations of GF and IM. Specifically, we found that GF and IM induced enlargement of the lysosomal compartment in leukemia K562 and HL-60 (Figure 1). Lysosomal sequestration of IM was greater than GF in both cell lines (Figure 1). This fact is understandable due to the physicochemical properties of both substances, the log P is 3.75 and 4.38 for GF and IM, respectively, and the pKa (strongest base) is 6.85 and 8.27 for GF and IM, respectively, according to the Drugbank. Regarding cell line differences, there was a slightly greater lysosomal accumulation of TKIs in K562 cells (Figure 1). In contrast to others [24,37], enlargement of the lysosomal compartment cannot be explained by lysosomal biogenesis in this case as there was no increased expression of lysosomal marker proteins in our experimental system (Figure 2, Figure 3 and Figure 4). In fact, we observed a decrease in hydrolase activity at high IM and GF concentrations of (Figure 4). While we have no clear explanation for this finding, it can be speculated that it may be a manifestation of exocytosis, where the lysosomal membrane proteins fuse with the plasma membrane (remaining part of the cell and hence there is no decrease in content) and soluble hydrolases are released into the extracellular space (enzymatic activity decreases). Indeed, recent observations have shown that lysosomal sequestration of weak base drugs is associated with lysosomal exocytosis [38].

Instead, we found that cell treatment with GF and IM induced changes in lysosomal morphology: while their size increased, their number decreased (Figure 5 and Figure 6). In addition, we observed that changes in lysosomal morphology can be prevented by the NAADP antagonist, *trans*-Ned 19 (Figure 7 and Figure 8), and by Ca^2+^ chelator, BABTA-AM (Figure 9 and Figure 10). These findings together suggested that both TKIs induced lysosomal fusion mediated by NAADP/Ca^2+^ signaling pathway [38,39]. Importantly, a theoretical analysis of our results clearly indicates that the lysosomal fusion itself represents an efficient mechanism generating the increased lysosomal sequestration capacity for weak base drugs (Figure 11).

Our results are in good agreement with published findings on the general function of lysosomes in basic cellular processes. It is now well documented that lysosomes serves as the terminal catabolic compartment of autophagy. They represent the central signaling hub in the regulation of cell metabolism, cholesterol homeostasis, membrane trafficking, the immune response, cell death, and tissue remodeling. These complex functions are largely under transcriptional control of TFEB in cooperation with lysosome-localized mTORC1, and related transcription factors, including microphthalmia transcription factor (MITF) and transcription factor E3 (TFE3) [25,26,39,40].

An increase in the lysosomal accumulation capacity may thus be an aspect of lysosomal biogenesis which combines both biosynthetic and membrane trafficking pathways. Apart from other functions, lysosomes serve as Ca^2+^ storage organelles which may on release, induce hetero or homotypic fusion. Ca^2+^ driven lysosomal membrane fusion with other lysosomes, lysosome related organelles, endosomes, or plasma membrane, regulate important cellular functions such as autophagy, endocytosis, exocytosis, and phagocytosis [41,42]. Although our results indicate homotypic fusion among lysosomes themselves (Figure 5 and Figure 6), we cannot exclude the possibility that the observed fusion includes heterotypic fusion i.e., fusion of lysosome(s) with endosome(s). This issue must be studied further. Similarly, it is unclear which ion channel was responsible for Ca^2+^ release, as inside the lysosomal membrane, there are several calcium channels that serve as NAADP receptors, including the two pore channels 1 and 2 (TPC1, 2) with high affinity for NAADP, and transient receptor potential melastatin 2 (TRPM2) with low affinity for NAADP [43]. Which of these channels are involved in NAADP-induced Ca^2+^ release has yet to be elucidated. We believe that our experimental model system is also well suited to identifying other components required for lysosomal fusion, i.e., membrane proteins and lipids that regulate their docking and subsequent fusion.

## 5. Conclusions

We demonstrated that extracellular TKIs, GF and IM, induced NAADP/Ca^2+^ mediated lysosomal fusion, leading to enlargement of the lysosomal compartment with a significantly increased sequestration capacity for these drugs without apparent lysosomal biogenesis.

## Figures and Tables

**Figure 1 biomolecules-10-00077-f001:**
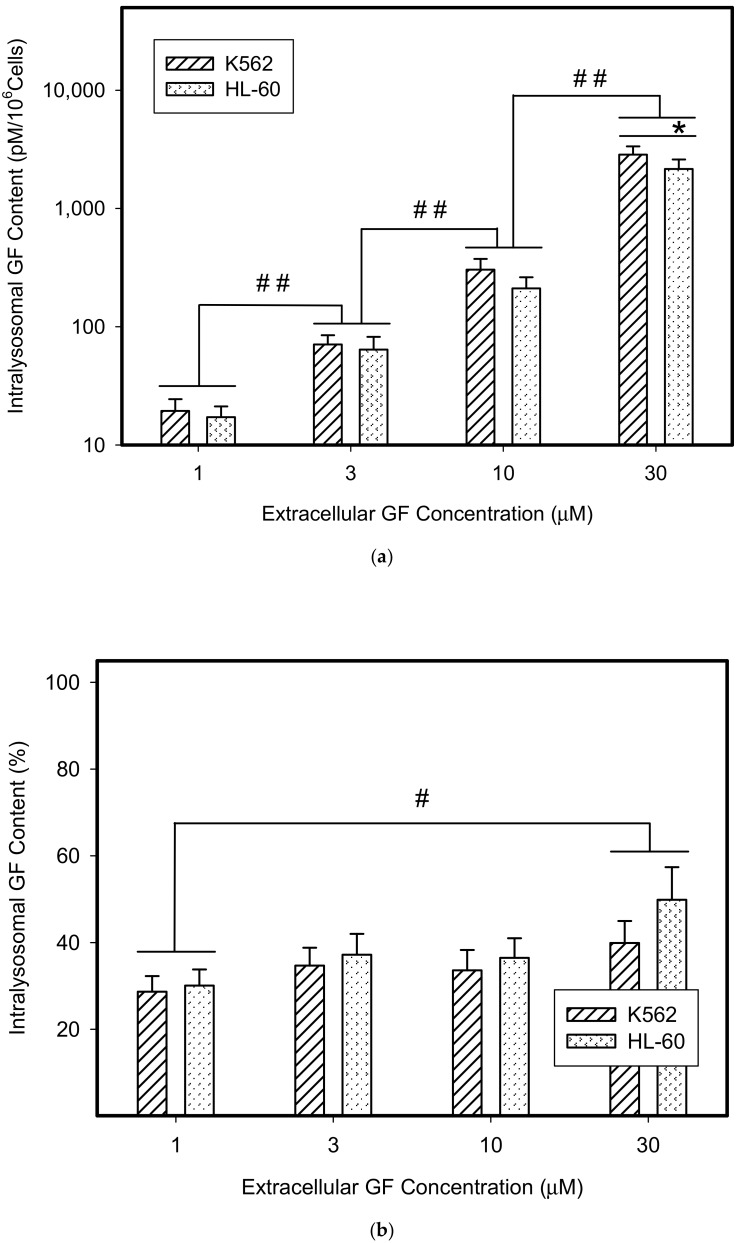
Lysosomal sequestration of tyrosine kinase inhibitors (TKIs). Absolute accumulation of TKI in lysosomes is expressed as molar amount of particular TKI in lysosomes per 10^6^ cells. Relative accumulation of TKIs is calculated as the ratio: (intralysosomal accumulation of particular TKI/intracellular accumulation of particular TKI) × 100%. (**a**) Absolute accumulation of gefitinib (GF) in lysosomes of cancer cells. (**b**) Relative accumulation of GF in lysosomes of cancer cells. (**c**) Absolute accumulation of imatinib (IM) in lysosomes of cancer cells. (**d**) Relative accumulation of IM in lysosomes of cancer cells. The columns represent the means of four independent experiments with standard deviations. * denotes significant change in the intralysosomal GF or IM content (*p* < 0.05) between the K562 and HL-60 cells. # denotes significant change in the intralysosomal content of GF or IM (*p* < 0.05) between the indicated groups. ## denotes a very significant change in the intralysosomal content of GF or IM (*p* < 0.01) between the indicated groups.

**Figure 2 biomolecules-10-00077-f002:**
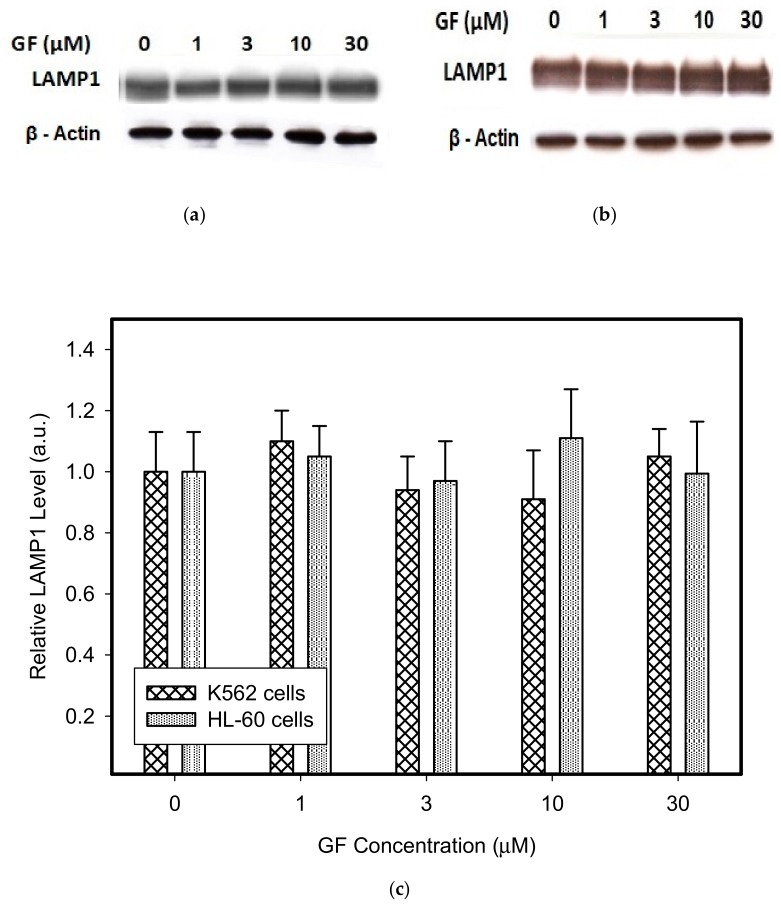
Effect of GF on expression of lysosomal proteins in cancer cells. Cells were cultured for 6 h in the presence of GF as indicated, prior to Western blot analysis. Cells cultured in medium without GF were taken as a control. (**a**) Western blot analysis of LAMP1 in K562 cells (typical analysis). (**b**) Western blot analysis of LAMP1 in HL-60 cells (typical analysis). (**c**) Quantitative analysis of LAMP1 expression using densitometry. (**d**) Western blot analysis of LAMP2 in K562 cells (typical analysis). (**e**) Western blot analysis of LAMP2 in HL-60 cells (typical analysis). (**f**) Quantitative analysis of LAMP2 expression using densitometry. (**g**) Western blot analysis of vacuolar ATPase subunit B2 in K562 cells (typical analysis). (**h**) Western blot analysis of vacuolar ATPase subunit B2 in HL-60 cells (typical analysis). (**i**) Quantitative analysis of vacuolar ATPase subunit B2 using densitometry. Results were normalized to β-actin. The results represent the means of four independent experiments with standard deviations.

**Figure 3 biomolecules-10-00077-f003:**
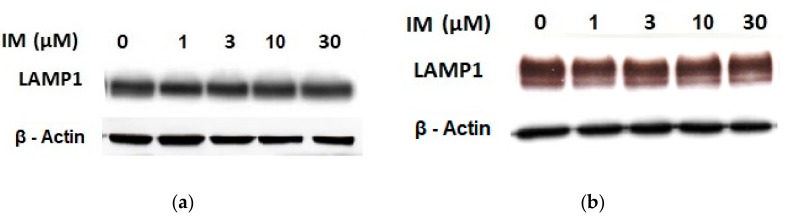
Effect of IM on expression of lysosomal proteins in cancer cells. Cells were cultured for 6 h in the presence of IM as indicated prior to Western blot analysis. Cells cultured in medium without IM were taken as a control. (**a**) Western blot analysis of LAMP1 in K562 cells (typical analysis). (**b**) Western blot analysis of LAMP1 in HL-60 cells (typical analysis). (**c**) Quantitative analysis of LAMP1 expression using densitometry. (**d**) Western blot analysis of LAMP2 in K562 cells (typical analysis). (**e**) Western blot analysis of LAMP2 in HL-60 cells (typical analysis). (**f**) Quantitative analysis of LAMP2 expression using densitometry. (**g**) Western blot analysis of vacuolar ATPase subunit B2 in K562 cells (typical analysis). (**h**) Western blot analysis of vacuolar ATPase subunit B2 in HL-60 cells (typical analysis). (**i**) Quantitative analysis of vacuolar ATPase subunit B2 using densitometry. Results were normalized to β-actin. The results represent the means of four independent experiments with standard deviations.

**Figure 4 biomolecules-10-00077-f004:**
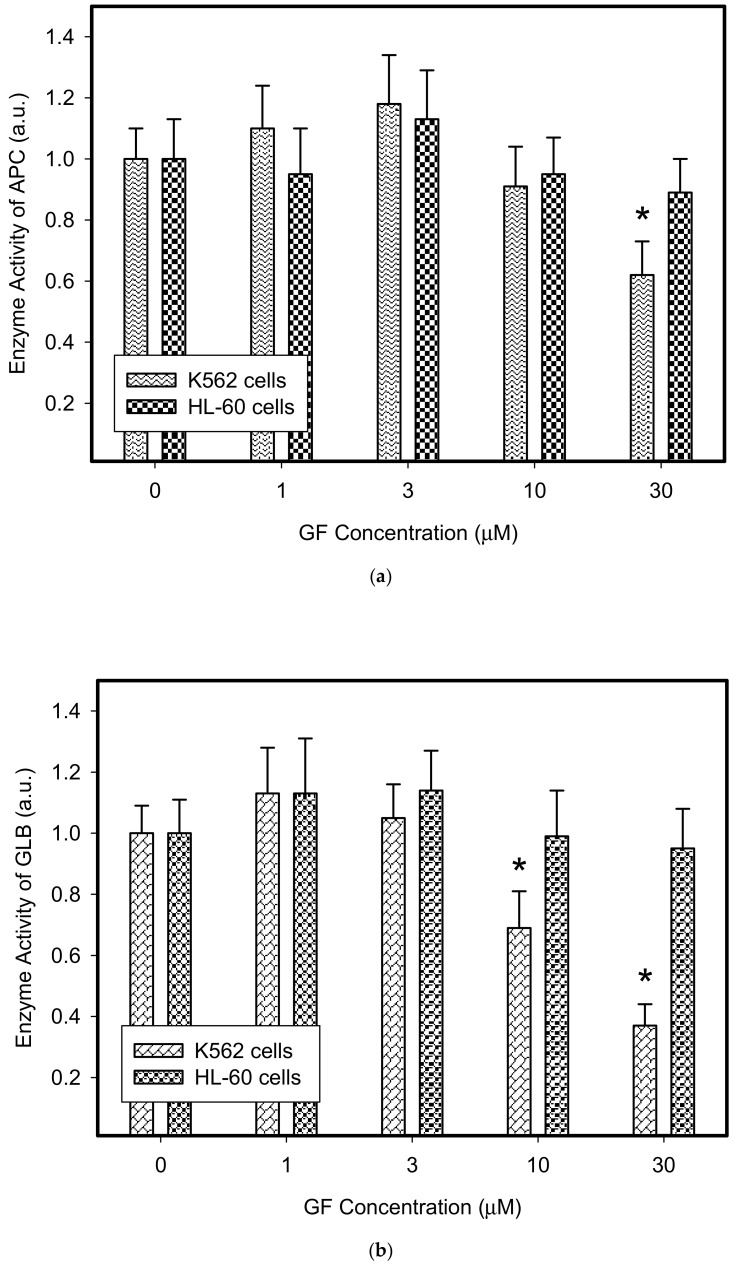
Effect of TKIs on activity of lysosomal hydrolases in cancer cells. Cells were cultured for 6 h in the presence of IM or GF as indicated and then enzymatic analysis of lysosomal hydrolases was done. Cells cultured in medium without TKIs were taken as a control. (**a**) Enzyme activity of lysosomal ACP in GF treated cells. (**b**) Enzyme activity of lysosomal GLB in GF treated cells. (**c**) Enzyme activity of lysosomal ACP in IM treated cells. (**d**) Enzyme activity of lysosomal GLB in IM treated cells. The columns represent the means of four independent experiments with standard deviations. * denotes significant change in the activity of lysosomal APC (or GLB) (*p* < 0.05) between TKI treated and control cells.

**Figure 5 biomolecules-10-00077-f005:**
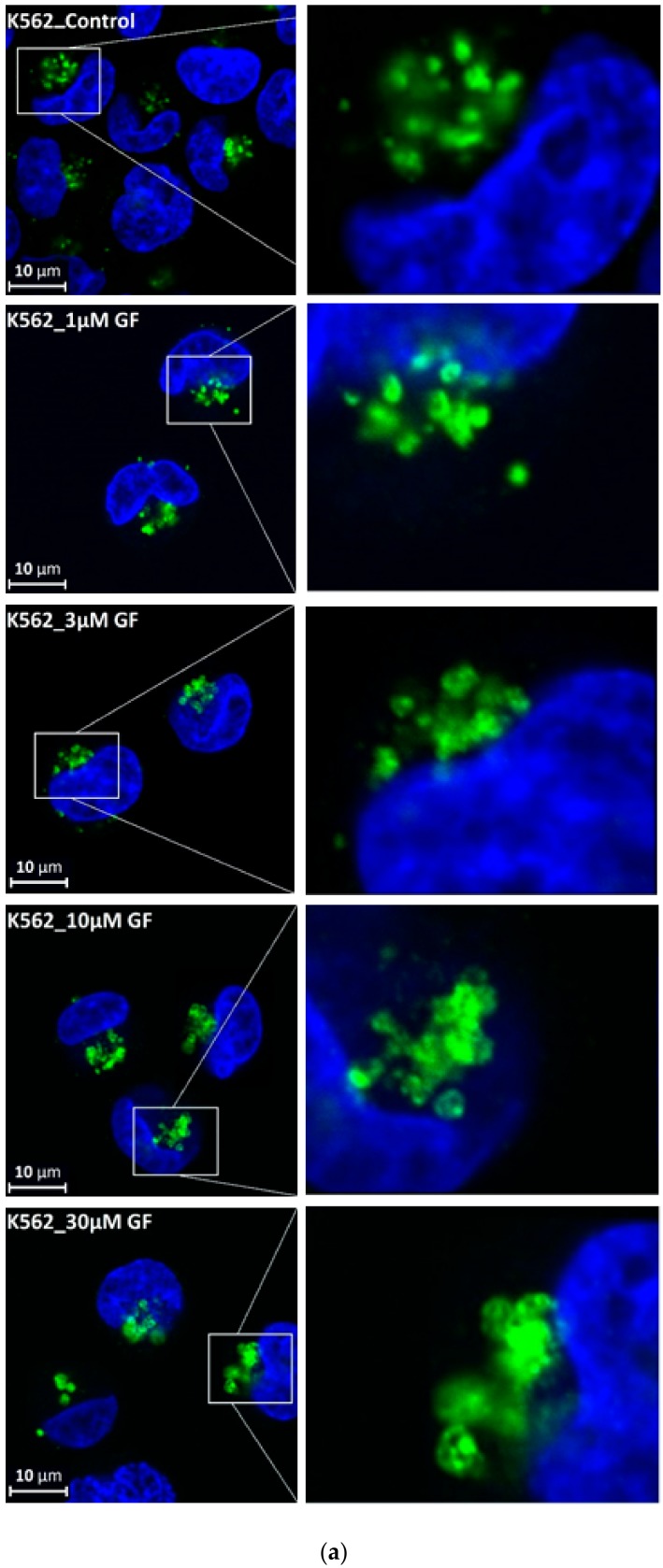
Effect of TKIs on lysosomal morphology in K562 cells. Cells were cultured for 6 h in the presence of particular TKI under standard conditions and then fixed in paraformaldehyde and subjected to the immunostaining of LAMP. Cells cultured in medium without TKIs were taken as a control. (**a**) Morphologic changes in lysosomes in IM treated cells (typical analysis). (**b**) Quantitative analysis of lysosomal size in IM treated cells. (**c**) Quantitative analysis of the number of lysosomes in IM treated cells. (**d**) Morphologic changes in lysosomes in GF treated cells (typical analysis). (**e**) Quantitative analysis of lysosomal size in GF treated cells. (**f**) Quantitative analysis of the number of lysosomes in GF treated cells. Lysosomal morphology was evaluated in at least 250 cells for each treatment. The results are represented as means of four independent experiments with standard deviations. ** denotes significant change in the relative number of lysosomes (*p* < 0.01) between TKI treated and control cells.

**Figure 6 biomolecules-10-00077-f006:**
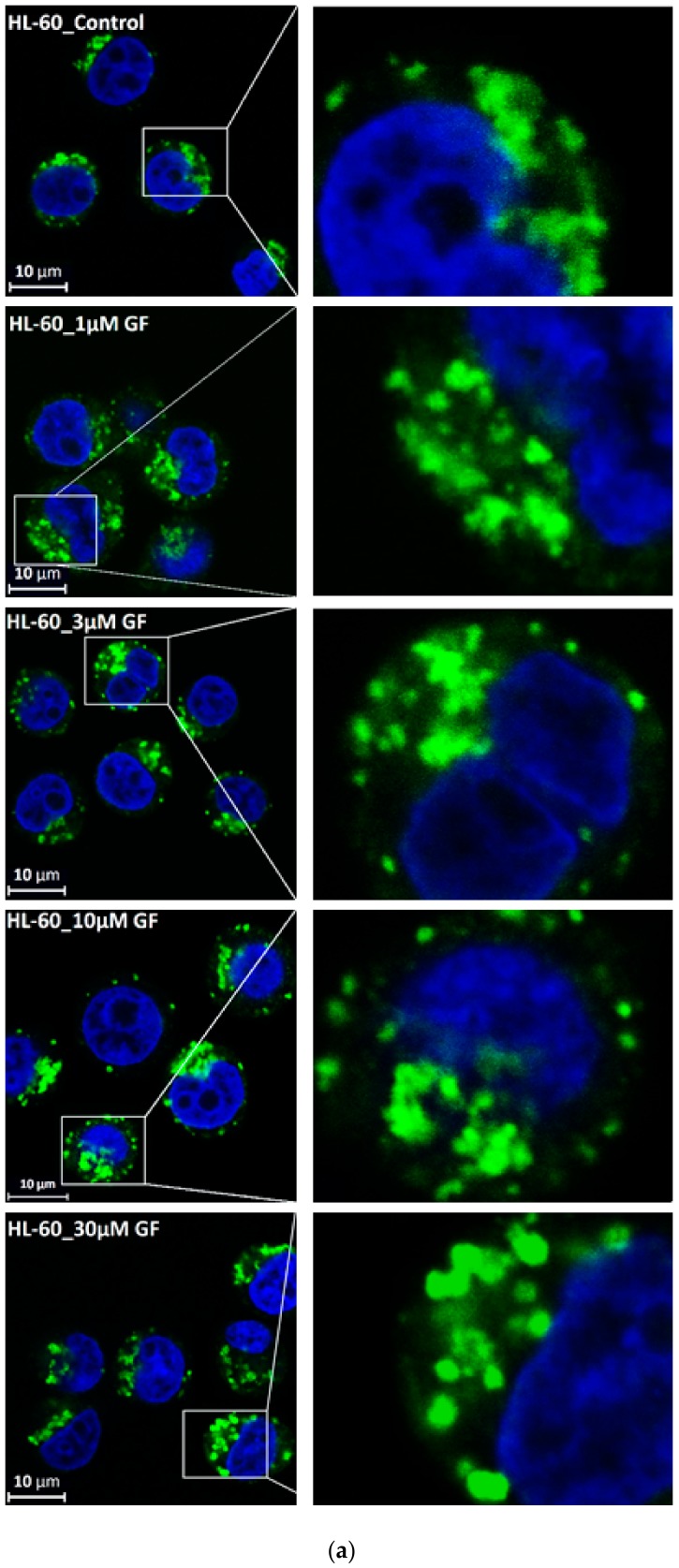
Effect of TKIs on lysosomal morphology in HL-60 cells. Cells were cultured for 6 h in the presence of particular TKI under standard conditions and then fixed in paraformaldehyde and subjected to the immunostaining of LAMP1. Cells cultured in medium without TKIs were taken as a control. (**a**) Morphologic changes in lysosomes in IM treated cells (typical analysis). (**b**) Quantitative analysis of lysosomal size in IM treated cells. (**c**) Quantitative analysis of the number of lysosomes in IM treated cells. (**d**) Morphologic changes in lysosomes in GF treated cells (typical analysis). (**e**) Quantitative analysis of lysosomal size in GF treated cells. (**f**) Quantitative analysis of the number of lysosomes in GF treated cells. Lysosomal morphology was evaluated in at least 250 cells for each treatment. The results are represented as means of four independent experiments with standard deviations. * denotes significant change in the relative number of lysosomes (*p <* 0.05) between TKI treated and control cells. ** denotes significant change in the relative number of lysosomes (*p <* 0.01) between TKI treated and control cells.

**Figure 7 biomolecules-10-00077-f007:**
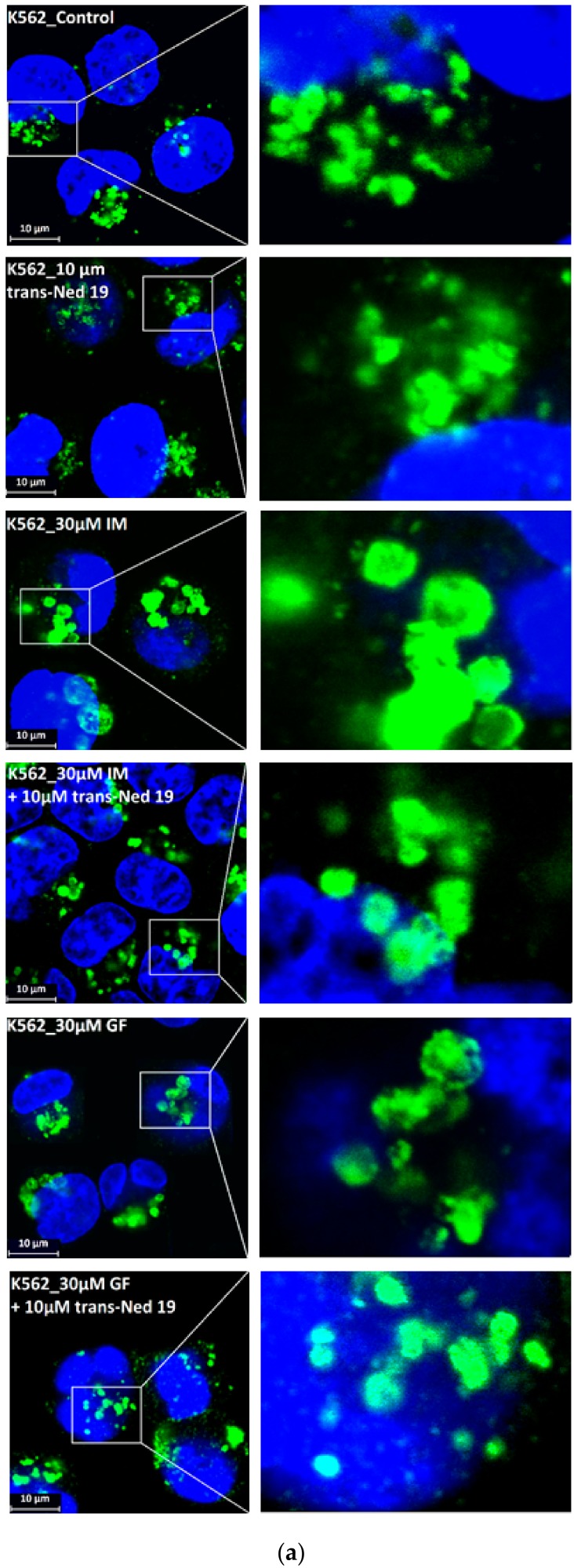
Effect of nicotinic acid adenine dinucleotide phosphate (NAADP) antagonist, *trans*-Ned 19, on lysosomal morphology in K562 cells treated with TKIs. Cells were cultured for 6 h in the presence of particular TKI with or without 10µM *trans*-Ned 19 under standard conditions and then fixed in paraformaldehyde and subjected to the immunostaining of LAMP1. Cells cultured in medium without TKIs were taken as a control. (**a**) Morphologic changes in lysosomes of K562 cells (typical analysis). (**b**) Quantitative analysis of lysosomal size in GF treated K562 cells. (**c**) Quantitative analysis of lysosomal size in IM treated K562 cells. Lysosomal morphology was evaluated in at least 250 cells for each treatment. The results are represented as means of four independent experiments with standard deviations. * denotes significant change in the number of lysosomes with size > 0.5µm (*p* < 0.05) between cells treated with TKI and cells treated with TKI + *trans*-Ned 19. ** denotes significant change in the number of lysosomes with size > 0.5µm (*p* < 0.01) between cells treated with TKI and cells treated with TKI + *trans*-Ned 19.

**Figure 8 biomolecules-10-00077-f008:**
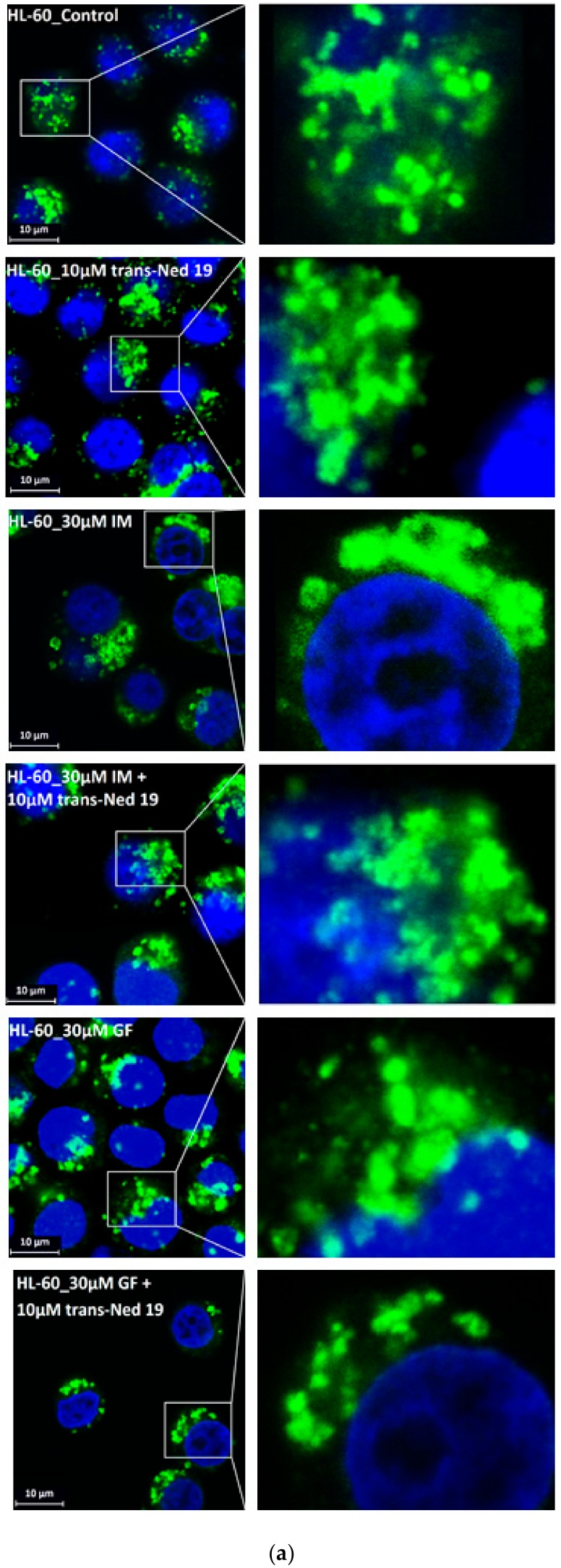
Effect of NAADP antagonist, *trans*-Ned 19, on lysosomal morphology in HL-60 cells treated with TKIs. Cells were cultured for 6 h in the presence of particular TKI with or without 10µM *trans*-Ned 19 under standard conditions and then fixed in paraformaldehyde and subjected to the immunostaining of LAMP1. Cells cultured in medium without TKIs were taken as a control. (**a**) Morphologic changes in lysosomes of HL-60 cells (typical analysis). (**b**) Quantitative analysis of lysosomal size in GF treated HL-60 cells. (**c**) Quantitative analysis of lysosomal size in IM treated HL-60 cells. Lysosomal morphology was evaluated in at least 250 cells for each treatment. Results are shown as means of four independent experiments with standard deviations. * denotes significant change in the number of lysosomes with size > 0.5µm (*p* < 0.05) between cells treated with TKI and cells treated with TKI + *trans*-Ned 19. ** denotes significant change in the number of lysosomes with size > 0.5µm (*p* < 0.01) between cells treated with TKI and cells treated with TKI + *trans*-Ned 19.

**Figure 9 biomolecules-10-00077-f009:**
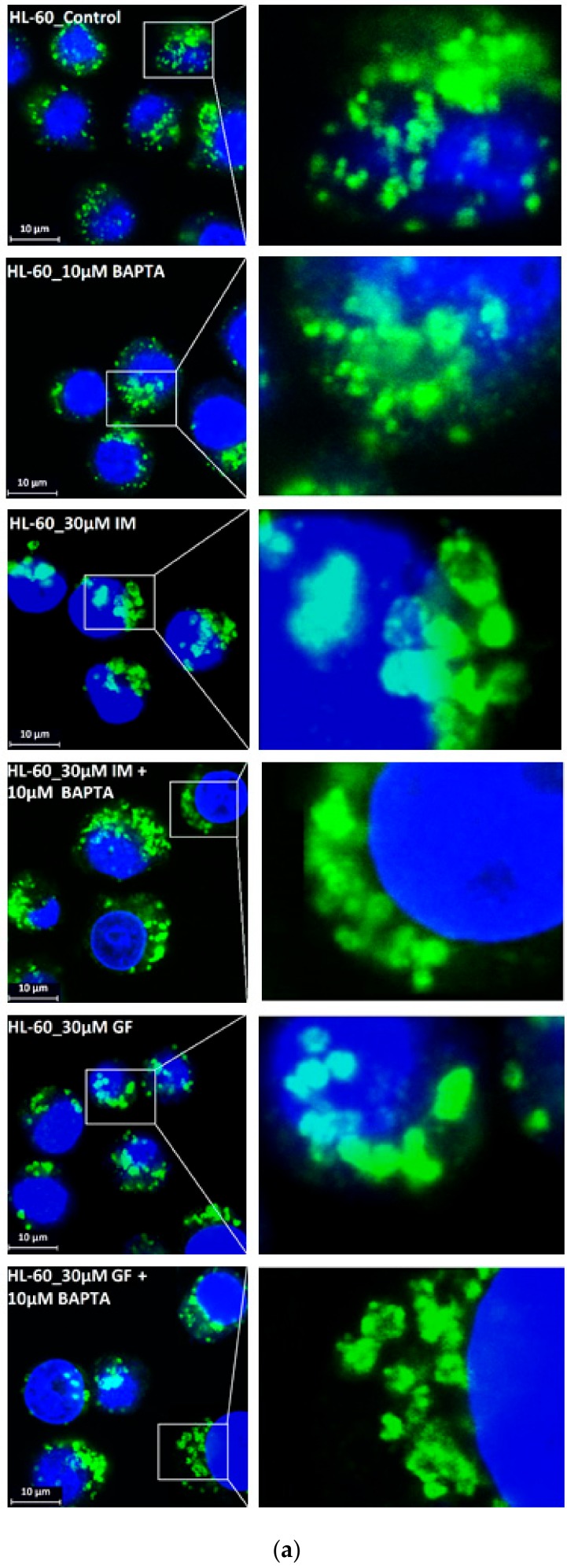
Effect of calcium chelator, BABTA-AM, on lysosomal morphology in K562 cells treated with TKIs. Cells were cultured for 6 h in the presence of particular TKI with or without 10µM BAPTA-AM under standard conditions and then fixed in paraformaldehyde and subjected to the immunostaining of LAMP1. Cells cultured in medium without TKIs were taken as a control. (**a**) Morphologic changes in lysosomes of K562 cells (typical analysis). (**b**) Quantitative analysis of lysosomal size in GF treated K562 cells. (**c**) Quantitative analysis of lysosomal size in IM treated K562 cells. Lysosomal morphology was evaluated in at least 250 cells for each treatment. The results are shown as means of four independent experiments with standard deviations. * denotes significant change in the number of lysosomes with size > 0.5µm (*p* < 0.05) between cells treated with TKI and cells treated with TKI + BAPTA-AM. ** denotes significant change in the number of lysosomes with size > 0.5µm (*p* < 0.01) between cells treated with TKI and cells treated with TKI + BAPTA-AM.

**Figure 10 biomolecules-10-00077-f010:**
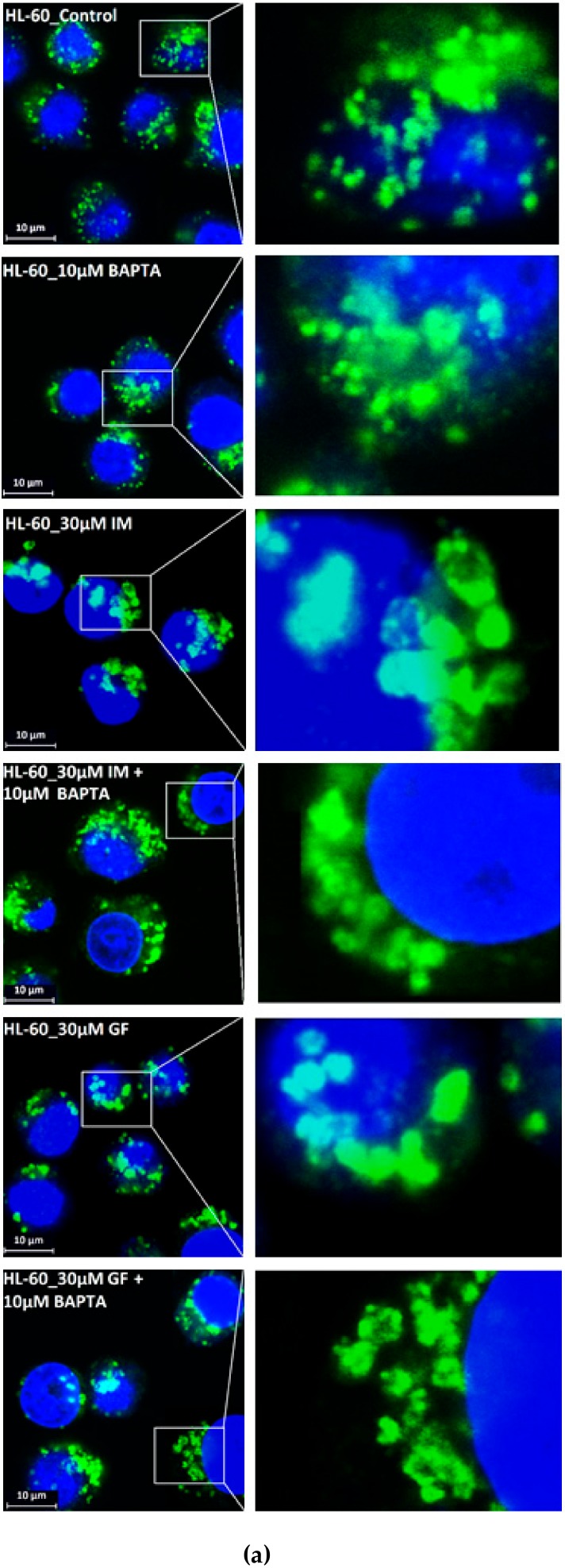
Effect of calcium chelator, BABTA-AM, on lysosomal morphology in HL-60 cells treated with TKIs. Cells were cultured for 6 h in the presence of particular TKI with or without 10 M BAPTA-AM under standard conditions and then fixed in paraformaldehyde and subjected to the immunostaining of LAMP1. Cells cultured in medium without TKIs were taken as a control. (**a**) Morphologic changes in lysosomes of HL-60 cells (typical analysis). (**b**) Quantitative analysis of lysosomal size in GF treated HL-60 cells. (**c**) Quantitative analysis of lysosomal size in IM treated HL-60 cells. Lysosomal morphology was evaluated in at least 250 cells for each treatment. The results are shown as means of four independent experiments with standard deviations. * denotes significant change in the number of lysosomes with size > 0.5 µm (*p* < 0.05) between cells treated with TKI and cells treated with TKI + BAPTA-AM. ** denotes significant change in the number of lysosomes with size > 0.5 µm (*p* < 0.01) between cells treated with TKI and cells treated with TKI + BAPTA-AM.

**Figure 11 biomolecules-10-00077-f011:**
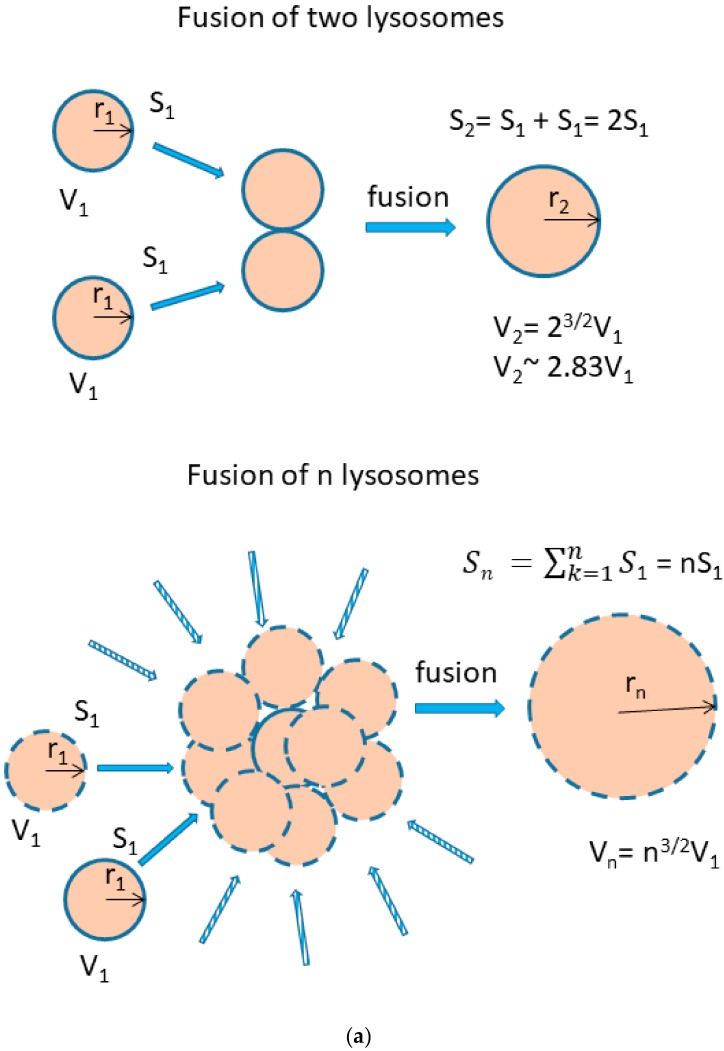
Lysosomal fusion and lysosomal volume. (**a**) Effect of lysosomal fusion on lysosomal volume. We consider a fusion of two lysosomes. For simplicity, they have the same size and they are spherical. Their lysosomal surface can be described by equation (I) and their volume by equation (II). S_1_ = 4πr_1_^2^ (I), V_1_ = 4/3πr_1_^3^ (II). The fusion of two lysosomes results in a new one whose membrane (surface) is formed by joining two membranes. The surface of the newly formed lysosome can be expressed by the equations (III and IV): S_2_ = 4πr_2_^2^ (III), S_2_ = 2S_1_ = 8πr_1_^2^ (IV). Combination III and IV gives: r_2_ = √2 r_1_ (V). The volume of the newly formed lysosome can be expressed by the equations: V_2_ = 4/3πr_2_^3^ (VI). Combination V and VI gives: V_2_ = 4/3π2^3^^/^^2^r_1_^3^ → V_2_ = 2.83V_1_. In general, when considering fusion of n lysosomes in one, the following relationship between surface and volume can be established: S_n_ = 4πr_n_^2^, S_n_ = nS_1_ = n4πr_1_^2^ → r_n_ = √n r_1_, V_n_ = 4/3πr_n_^3^, V_n_ = 4/3πn^3/2^r_1_^3^ → V_n_ = n^3/2^V_1_. (**b**) The functional relationship between the number of fusing lysosomes and resulting volume.

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
