# Peer review of "Lysosomal Fusion: An Efficient Mechanism Increasing Their Sequestration Capacity for Weak Base Drugs without Apparent Lysosomal Biogenesis"

_biomolecules, 2020, doi:10.3390/biom10010077_

Round 1

Reviewer 1 Report

Comments to the Author and Editor. Biomolecules-626192

An interesting article that aims to deepen into the mechanisms of lysosomes to sequester drugs. Although some of the results and the final conclusion do not convince me at all. During review the following concerns arose:

In the Instruction, I miss that the authors talk about the previous data that exist in the literature of the effect of tyrosine kinase inhibitors, gefitinib and imatinib, in the endocytic system and lysosomes. The authors should include some phrase about it.

Methods

As for the described method of Calculation of TKIs in lysosomes. The authors must justify the method used, since they affirm that the quantifications of GF and IM are in the lysosomal compartment. It would not be more correct for the authors to isolate the lysosomes by a preparative method and then measure the amount of GF and IM in these lysosomes (Methods Mol Biol. 2016;1449:299-311. doi: 10.1007/978-1-4939-3756-1_19; Curr Protoc Cell Biol. 2001 May;Chapter 3:Unit 3.6. doi: 10.1002/0471143030.cb0306s07).

In the Western blot analysis (pag. 5), figure the following text “Bcr-Abl signaling was analyzed using rabbit polyclonal anti-phospho-Bcr (Tyr177) antibody (1:1,000) and rabbit polyclonal anti-phospho-CrkL (Tyr207) antibody ((1:1000); Cell Signaling Technology, Denvers, MA)”. These results are not listed in the manuscript.

Regarding the measurement of the Activity of lysosomal hydrolases, I think that the most correct thing would be to do enzymatic activities in vivo. One that the cells has lysed, the environment inside the lysosomes has broken. Then the authors put a buffer with a pH that should be the one inside the lysosome. But we do not know if that pH is the real one after treating the cells with GF and IM. From what I believe, the authors should measure the activity of one of the lysosomal enzymes in vivo as described in (Int J Mol Sci. 2016;17:404. doi: 10.3390/ijms17030404).

Results

The authors do not justify why they have chosen these cell lines, human leukemia K562 and HL60.

Figure legend 2 and 3. Please, check the figures legends because it does not match the letters in the figures. Also indicate which western-blot corresponds to each cell line.

The authors do not indicate the number of independent experiments performed.

I suppose that when they indicate relative expression level, they have divided the quantification of the expression of the analyzed protein by that of beta-actin. Indicate in the figure legend that has been made.

As for the results of lysosomal enzymatic activities, I have already expressed my concern about lyse the cells and broken the lysosomes that could have changed the conditions inside them after treatment with GF and IM. Since the accumulation of these drugs inside the lysosome may be altered its pH, and it must be taken into account that the activity of lysosomal enzymes has a very determined pH of action. The malfunction of the enzymes causes certain material, that should be hydrolyzed, to accumulate inside lysosomal because it cannot be hydrolyzed, causing the lysosome to increase in size. I think it is best to measure the activity of the enzyme in vivo and in the case of GFB, it is possible to do it (Int J Mol Sci. 2016;17:404. doi: 10.3390/ijms17030404).

In most of the photos chosen in Figures 5 to 8, the enlarged control cell chosen is not the representative one. In fact, if you look at the less enlarged image, you realize that it is not the most representative. It seems that the authors have chosen the cell with more lysosomes. And in my experience, in condition of rest or control without treatment, the usual thing is that they have less lysosomes. Based on this fact, how many cells have been quantified for each condition? Because I believe that if the control cells that have been quantified are the cells with more labeling, the results could have been altered.

In reference to the theoretical analysis, I think it should be included in the methods section. Of course, in my case I am not an expert in this regard. But I see that they affirm many things without any bibliographic support.

For the discussion and final conclusion, after reviewing the results, it should be rewritten.

Author Response

Reviewer #1:

We thank the Reviewer for careful reading of our MS. We are delighted that you consider our article to be interesting. The MS was rewritten mostly according to your suggestions. In addition, we extended our results about the possible involvement of nicotinic acid adenine dinucleotide phosphate (NAADP) in the process of lysosomal fusion. Changes are in green.

Comments to the Author and Editor. Biomolecules-626192

An interesting article that aims to deepen into the mechanisms of lysosomes to sequester drugs. Although some of the results and the final conclusion do not convince me at all. During review the following concerns arose:

In the Instruction, I miss that the authors talk about the previous data that exist in the literature of the effect of tyrosine kinase inhibitors, gefitinib and imatinib, in the endocytic system and lysosomes. The authors should include some phrase about it.

Our answer:

According to your suggestions we added some information and references about the issue you mentioned. Please, see revised version of our MS, section Introduction, page 2, 4th paragraph.

Methods

As for the described method of Calculation of TKIs in lysosomes. The authors must justify the method used, since they affirm that the quantifications of GF and IM are in the lysosomal compartment. It would not be more correct for the authors to isolate the lysosomes by a preparative method and then measure the amount of GF and IM in these lysosomes (Methods Mol Biol. 2016;1449:299-311. doi: 10.1007/978-1-4939-3756-1_19; Curr Protoc Cell Biol. 2001 May;Chapter 3:Unit 3.6. doi: 10.1002/0471143030.cb0306s07).

Our answer:

The method you suggest is very good, but very laborious. However, even this method does not provide a completely pure fraction of lysosomes.In contrast, our method is less laborious and provides very reliable results. This method was originally described by Burger et al., Mol. Pharmacol. 2015, 88(3), 477-487, https://doi.org/10.1124/mol.114.097451 (This reference was mentioned in the original version of our MS). The method relies on the fact that selective inhibitors of vacuolar ATPase such as Bafilomycin A1 or Concanamycin A completely prevent lysosomal accumulation of the weak base drugs (Ouar et al., Biochem J 2003, 370: 185-193, Adar et al., Cell Death Dis. 2012, 3:e293, Zhitomirski and Assaraf Oncotarget 2015, 6, 1143-1156). As far as we know, the principles of this method have never been questioned.  

In the Western blot analysis (pag. 5), figure the following text “Bcr-Abl signaling was analyzed using rabbit polyclonal anti-phospho-Bcr (Tyr177) antibody (1:1,000) and rabbit polyclonal anti-phospho-CrkL (Tyr207) antibody ((1:1000); Cell Signaling Technology, Denvers, MA)”. These results are not listed in the manuscript.

Our answer:

This was a mistake. This text was deleted. Please, see revised version of our MS, section Materials and Methods, page 4, subsection 2.6. Western blot analysis.

Regarding the measurement of the Activity of lysosomal hydrolases, I think that the most correct thing would be to do enzymatic activities in vivo. One that the cells has lysed, the environment inside the lysosomes has broken. Then the authors put a buffer with a pH that should be the one inside the lysosome. But we do not know if that pH is the real one after treating the cells with GF and IM. From what I believe, the authors should measure the activity of one of the lysosomal enzymes in vivo as described in (Int J Mol Sci. 2016;17:404. doi: 10.3390/ijms17030404).

Our answer:

At this point we do not agree with the Reviewer since we did not want to measure the actual activity of lysosomal hydrolases but their content i.e., their expression levels. Therefore, we could not use the smart protocol you mentioned.

The in situ method you suggested is brilliant for the measurement of different effects on activity of lysosomal hydrolases such as the presence of drugs and/or changed pH can affect activity of lysosomal hydrolases. However, this is not our case.

Nevertheless, we add some text to clarify application of our method. Please, see revised version of our MS, section Materials and Methods, page 5, subsection 2.7. Activity of lysosomal hydrolases

Results

The authors do not justify why they have chosen these cell lines, human leukemia K562 and HL60.

Our answer:

According to your suggestion this information was added. Please, see revised version of our MS, section Results, page 6, 1st paragraph.

Figure legend 2 and 3. Please, check the figures legends because it does not match the letters in the figures. Also indicate which western-blot corresponds to each cell line.

Our answer:

We improve the figure legends according to your suggestion. Please, see revised version of our MS, section Legends to Figures, page 9, Fig. 2 and Fig. 3.

The authors do not indicate the number of independent experiments performed.

Our answer:

Results represent the means of four independent experiments with SD. This information was added according to your suggestion. Please, see revised version of our MS, section Legends to Figures, pages 8-11.

I suppose that when they indicate relative expression level, they have divided the quantification of the expression of the analyzed protein by that of beta-actin. Indicate in the figure legend that has been made.

Our answer:

It was done as you wrote. So that, we add this information according to your suggestion. Please, see revised version of our MS, section Legends to Figures, pages 8-9, Fig. 2 and Fig. 3.

As for the results of lysosomal enzymatic activities, I have already expressed my concern about lyse the cells and broken the lysosomes that could have changed the conditions inside them after treatment with GF and IM. Since the accumulation of these drugs inside the lysosome may be altered its pH, and it must be taken into account that the activity of lysosomal enzymes has a very determined pH of action. The malfunction of the enzymes causes certain material, that should be hydrolyzed, to accumulate inside lysosomal because it cannot be hydrolyzed, causing the lysosome to increase in size. I think it is best to measure the activity of the enzyme in vivo and in the case of GFB, it is possible to do it (Int J Mol Sci. 2016;17:404. doi: 10.3390/ijms17030404).

Our answer:

As mentioned above, at this point we do not agree with the Reviewer since we did not want to measure the actual activity of lysosomal hydrolases but their content i.e., their expression levels. Therefore, we could not use the smart protocol you mentioned.

In most of the photos chosen in Figures 5 to 8, the enlarged control cell chosen is not the representative one. In fact, if you look at the less enlarged image, you realize that it is not the most representative. It seems that the authors have chosen the cell with more lysosomes. And in my experience, in condition of rest or control without treatment, the usual thing is that they have less lysosomes. Based on this fact, how many cells have been quantified for each condition? Because I believe that if the control cells that have been quantified are the cells with more labeling, the results could have been altered.

Our answer:

We used other images which look more representative. In these experiments, 250 cells were evaluated for each treatment. We believe that this amount is representative. This is now stated in the text. Please, see revised version of our MS, section Legends to Figures, pages 9-11, Figs. 5 -10.

In reference to the theoretical analysis, I think it should be included in the methods section. Of course, in my case I am not an expert in this regard. But I see that they affirm many things without any bibliographic support.

Our answer:

Theoretical analysis is one of the results of our research. It provides clear evidence that fusion of existing lysosomes can "generate" an increase in lysosome volume capacity without biogenesis. Therefore, we leave it in the Results section. High school mathematics was used here, specifically formulas for calculating the surface and volume of a sphere. Therefore, we do not consider it necessary to give any literary references.

Reviewer 2 Report

The research presented here adds to the understanding the cellular mechanism of lysosomal sequestration of the clinically important drugs gefitinib and imatinib. However, there are a couple of deficiencies in the description of the methods and results.

1) There is no indication of statistical significance in figures 1b and 1c to support the finding of an increased lysosomal accumulation capacity.  Does this indicate that there is no statistically significant difference or has the annotation been omitted? If there is no statistically significant difference could the authors please explain why they consider the difference to be physiologically important?

2) There is no description of how the  quantitative analysis of morphological changes in lysosomes of cells was performed.

The images appear to be captured on a wide field fluorescence microscope. Is the resolution of the images adequate for accurate measurement of the lysosomes morphology? 

The authors should include details of the microscope and objective lenses and digital camera used for these experiments.

How was the three dimensional size and shape of the lysosomes measured?

What image analysis sofware was used?

How many cells/fields of view were analysed for each experiment?

Author Response

Reviewer #2:

We thank the Reviewer for careful reading of our MS. The MS has been supplemented with the data you suggested. In addition, we extended our results about the possible involvement of nicotinic acid adenine dinucleotide phosphate (NAADP) in the process of lysosomal fusion. Changes are in red.

The research presented here adds to the understanding the cellular mechanism of lysosomal sequestration of the clinically important drugs gefitinib and imatinib. However, there are a couple of deficiencies in the description of the methods and results.

1) There is no indication of statistical significance in figures 1b and 1c to support the finding of an increased lysosomal accumulation capacity.  Does this indicate that there is no statistically significant difference or has the annotation been omitted? If there is no statistically significant difference could the authors please explain why they consider the difference to be physiologically important?

Our answer:

There is a significant increase in lysosomal accumulation capacity. According to your suggestion we added this information. Please, see revised version of our MS, pages 15-16, Fig. 1.  

2) There is no description of how the quantitative analysis of morphological changes in lysosomes of cells was performed.

Our answer:

The quantitative analysis was performed using a confocal microscopy, where each cell was screened in several layers and lysosomes were then counted within whole cell. In this way, lysosomes were counted in total 250 cells. The morphological changes were then analysed from these images by ImageJ software.

3) The images appear to be captured on a wide field fluorescence microscope. Is the resolution of the images adequate for accurate measurement of the lysosomes morphology? 

Our answer:

In our set-up imaging was performed using confocal microscopy, for the objective 100x with NA 1.4 the resolution of 0,20 µm for 550 nm is even increased for wavelength 488 nm to ca 0,17 µm. Which makes possible visualization of lysosomes which size varies from 0.3 to 2.5 µm. We are aware that for excitation by 488 nm laser the real size of an airy disc for Olympus 100x objective NA 1,4 is 4,1 mm and optimal pinhole size represents 50-60% of this value. (Pawley, J.: Handbook of Biological Confocal Microscopy). Size of pinhole was set to collect enough of FITC signal (specified in MS text) and images exported with increased channel intensity (the same algorithm applied to all images) for better presentation of data.

4) The authors should include details of the microscope and objective lenses and digital camera used for these experiments.

Our answer:

Details were provided in revised version of our MS, section Materials and Methods, pages 5-6. Signal from confocal microscope is not recorded by CCD camera like in fluorescence microscopes but in each channel the signal point detection is recorded by confocal unit and visualized in sw.

5) How was the three dimensional size and shape of the lysosomes measured?

Our answer:

Lysosomes are vesicles derived from ER. Therefore, it is natural to consider them as spheres in 3D and circles in 2D, even if they do not have a perfect geometric shape. And so they also appeared in the images. It is unthinkable to consider them spheroids, cubes, or pyramids. When analyzing their size, we measured their diameters.

6) What image analysis sofware was used?

Our answer:

We used the ImageJ software (free software provided by NIH - National Institutes of Health; Wayne Rasband)

7) How many cells/fields of view were analysed for each experiment?

Our answer:

For each experiment were analysed in total 250 cells.

Round 2

Reviewer 1 Report

As it appears in the manuscript pag 5.

“2.7. Activity of lysosomal hydrolases

The enzymatic reaction was initiated by adding cell extract (equivalent of 100 μg protein) to the cell assay buffer (50mM NaCl, 50mM citrate–phosphate buffer, pH 4.5) containing appropriate substrate. ACP activity was measured using 1mM 4-methylumbelliferyl phosphate as a substrate. GLB activity was measured using 1mM 4-methylumbelliferyl-β-D-galactopyranoside as a substrate. The reaction mixture was incubated at 37 C for 30 min and then the enzymatic reaction was stopped by adding Tris buffer, pH 10.7. The relative fluorescence of released 4-methylumbelliferone was monitored at 365/445nm.”

This is the method that appears in the paper, what they measured is an enzymatic activity, not the expression how they want to make me believe with their answers. For each enzyme they put a substrate that then measure the restoring product of the enzymatic activity by means of the fluorescence that it emits. That a protein is more expressed does not have to coincide with more enzymatic activity. And I insist again on the above in the previous review.

Regarding the measurement of the Activity of lysosomal hydrolases, I think that the most correct thing would be to do enzymatic activities in vivo. One that the cells has lysed, the environment inside the lysosomes has broken. Then the authors put a buffer with a pH that should be the one inside the lysosome. But we do not know if that pH is the real one after treating the cells with GF and IM. From what I believe, the authors should measure the activity of one of the lysosomal enzymes in vivo as described in (Int J Mol Sci. 2016;17:404. doi: 10.3390/ijms17030404).

As for the results of lysosomal enzymatic activities, I have already expressed my concern about lyse the cells and broken the lysosomes that could have changed the conditions inside them after treatment with GF and IM. Since the accumulation of these drugs inside the lysosome may be altered its pH, and it must be taken into account that the activity of lysosomal enzymes has a very determined pH of action. The malfunction of the enzymes causes certain material, that should be hydrolyzed, to accumulate inside lysosomal because it cannot be hydrolyzed, causing the lysosome to increase in size. I think it is best to measure the activity of the enzyme in vivo and in the case of GFB, it is possible to do it (Int J Mol Sci. 2016;17:404. doi: 10.3390/ijms17030404).

Author Response

Dear Reviewer,

Unfortunately, we can't agree with you.

Generally, enzymatic activity and expression level may not always correlate. However, enzymatic activity and expression levels (= amount of enzyme) usually correlate. After all, this teaches us biochemistry textbooks for universities.

Measurement of lysosomal hydrolase activity in cell lysates reflects the amount of enzyme in lysosomes (protein amount = expression). And we wanted to measure it. That's why we did it.

Instead, you want us to measure hydrolase activity in vivo (but in fact, you mean in situ). The measurement you propose is amazing, but inappropriate for our purpose. We did not want to measure the effect of IM and GF on particular hydrolase activity in situ! In this measurement, the observed activity of a particular hydrolase reflects the influence of a number of factors (e.g., changes in lysosomal pH, direct effect of IM and GF to modulate the activity of a particular hydrolase, changes in expression of a particular enzyme, etc.). The results obtained in this way can hardly be interpreted without further measurements! Therefore, we consider your procedure less suitable for our purposes.

Importantly, significantly elevated enzyme activity of beta-hexosaminidase in cell extracts(!) was used as evidence of lysosomal biogenesis in the paper by Zhitomirski and Assaraf Oncotarget. 2015, 6, 1143-1156.

Interestingly, we used the same approach to demonstrate that there is NO lysosomal biogenesis because we did not find any significant increase in enzyme activity of TWO hydrolases. However, in the case of our MS is this approach suddenly wrong or insufficient.